# C3 complement inhibition prevents antibody-mediated rejection and prolongs renal allograft survival in sensitized non-human primates

Robin Schmitz [1], Zachary W. Fitch [1], Paul M. Schroder[1], Ashley Y. Choi [1], Miriam Manook [1], Janghoon Yoon [1], Mingqing Song [1], John S. Yi [2], Sanjay Khandelwal [3], Gowthami M. Arepally[3], Alton B. Farris [4], Edimara S. Reis[5], John D. Lambris[5], Jean Kwun [1,6✉] & Stuart J. Knechtle [1,6✉]

Sensitized kidney transplant recipients experience high rates of antibody-mediated rejection due to the presence of donor-specific antibodies and immunologic memory. Here we show that transient peri-transplant treatment with the central complement component C3 inhibitor Cp40 significantly prolongs median allograft survival in a sensitized nonhuman primate model. Despite donor-specific antibody levels remaining high, fifty percent of Cp40-treated primates maintain normal kidney function beyond the last day of treatment. Interestingly, presence of antibodies of the IgM class associates with reduced median graft survival (8 vs. 40 days; $p = 0.02$). Cp40 does not alter lymphocyte depletion by rhesus-specific anti-thymocyte globulin, but inhibits lymphocyte activation and proliferation, resulting in reduced antibody-mediated injury and complement deposition. In summary, Cp40 prevents acute antibody-mediated rejection and prolongs graft survival in primates, and inhibits T and B cell activation and proliferation, suggesting an immunomodulatory effect beyond its direct impact on antibody-mediated injury.

[1] Duke Transplant Center, Department of Surgery, Duke University School of Medicine, Durham, NC, USA. [2] Division of Surgical Sciences, Department of Surgery, Duke University, Durham, NC, USA. [3] Department of Medicine, Duke University School of Medicine, Durham, NC, USA. [4] Department of Pathology, Emory University School of Medicine, Atlanta, GA, USA. [5] Department of Pathology and Laboratory Medicine, University of Pennsylvania, Philadelphia, PA, USA. [6] These authors contributed equally: Jean Kwun, Stuart J. Knechtle. ✉email: jean.kwun@duke.edu; stuart.knechtle@duke.edu

Kidney transplantation is the treatment of choice for patients with end-stage renal disease. Compared to dialysis, it improves quality of life and reduces long-term mortality and cost of care[1–4]. However, kidney transplant candidates who are sensitized to a wide variety of HLA antigens from prior blood transfusions, pregnancies, or allograft failure have greatly reduced chances of finding an HLA-compatible donor. While the proportion of waitlisted candidates with calculated panel-reactive antibody (cPRA) of 98–100% is declining since the introduction of the new kidney allocation system in 2014, they still represent a disproportionate 7.2% of the wait list in the most recent OPTN/SRTR report[5]. Additionally, ~25% of candidates have a cPRA between 20 and 98%[5], and when transplanted these recipients have higher rates of acute antibody-mediated rejection (AMR) and shorter graft survival due to the presence of pre-formed anti-HLA antibodies and immunologic memory[6–9].

In order to overcome this immunologic barrier to transplantation, strategies targeting donor-specific antibodies (DSA) of sensitized recipients prior to transplantation have evolved to enable HLA-incompatible transplantation[10]. Desensitization regimens aim to decrease DSA through plasmapheresis or immunoadsorption in combination with intravenous immunoglobulin (IVIg) infusions and depletion of B cells with rituximab[11]. However, due to the inability to deplete bone marrow–resident, terminally differentiated plasma cells as well as memory B cells, these regimens are associated with AMR in up to 40% of patients, caused by a rebound of DSA after kidney transplantation[12]. Therefore, effective desensitization remains elusive for sensitized patients.

AMR is a clinical and histopathologic diagnosis based on clinical allograft dysfunction reflected by a decline in eGFR, histologic evidence of endothelial inflammation and associated features, and the presence of DSA in blood[13]. The primary mechanism by which DSAs cause AMR is through activation of the complement cascade by complement-fixing antibodies[14]. The complement-dependent cytotoxicity (CDC) crossmatch was for decades the gold standard to assess donor/recipient compatibility prior to transplantation[15]. In the 1990s Feucht et al. described the association of rejection and the complement split product C4d[16,17], which was subsequently confirmed to be a biomarker of AMR and added to the Banff criteria in the early 2000s[18]. Furthermore, complement contributes directly to activation of both the innate and adaptive immune systems as well as causing direct damage to the allograft[14,19].

Complement inhibition has been used in the field of transplantation over the last decade to prevent acute AMR in patients with high levels of complement-fixing DSA[7]. C1 inhibition has been tested in multiple pilot studies in combination with antibody-reducing regimens. C1 inhibition reduces ischemia-reperfusion injury, may prevent early AMR by abrogating the classical pathway activation mechanism[20], and improves the outcome of acute renal transplant AMR when combined with plasmapheresis and/or IVIg[21,22]. Eculizumab, a monoclonal antibody to C5, is directed at terminal complement pathways and prevents the formation of the membrane attack complex (MAC), and it has been used for the treatment of AMR in several solid organ transplants, including kidney[23], lung[24], and intestine[25]. In a trial that included 26 sensitized kidney transplant recipients, eculizumab was given in combination with standard-of-care antibody-reducing treatment prior to kidney transplantation. Eculizumab prophylaxis was shown to reduce the rate of acute rejection from 42.2 to 7.7% in the first 3 months after kidney transplantation[26].

The Cp40 family includes peptidic C3 inhibitors that act on native C3 and prevent its activation. Cp40, originally discovered in 1996, is a cyclic peptide that blocks convertase-mediated activation of C3 by all pathways[27–29]. Cp40 has been tested extensively in pre-clinical nonhuman primate (NHP) models to investigate toxicology and pharmacokinetics and shows good safety and a favorable pharmacokinetic profile. In vitro, Cp40 has shown its ability to abrogate the detrimental thrombo-inflammatory consequences of complement activation, as in ex vivo porcine-to-human models of xenotransplantation and xenoantibody-mediated complement cytotoxicity[30–32]. AMY-101, a derivative of the compstatin family, was evaluated in the first-in-human (FIH) clinical trial in healthy male volunteers, in which AMY-101 was shown to be safe and well tolerated (NCT03316521). Currently, AMY-101 is being evaluated to assess its safety and efficacy in adults with gingivitis (NCT03694444) and in acute respiratory distress syndrome (ARDS) due to COVID-19 (NCT04395456).

Here we show that the complement inhibitor compstatin (Cp40 or Cp40-KK, both referred to as Cp40), which targets the central complement component C3, prevents acute AMR and prolongs graft survival without the addition of antibody-reducing desensitization treatment in a highly sensitized NHP model.

## Results

**Cp40 prevents early antibody-mediated rejection and significantly prolongs graft survival in the presence of high levels of donor-specific antibodies.** Maximally MHC-mismatched NHP pairs were sensitized to each other with two sequential skin transplants. Five primates were assigned to the control group and 6 primates enrolled in the treatment group. MHC genotyping for all pairs is summarized in Supplementary Table 1. Data related to control animals were partially reported previously[33]. Successful sensitization was quantified by weekly flow cross-match, shown in Supplementary Fig. 1A. Both groups were equally sensitized with no significant difference in peak DSA levels prior to kidney transplantation (Supplementary Fig. 1B). All primates received induction therapy with rhesus-specific anti-thymocyte globulin (rhATG). Maintenance immunosuppression consisted of tacrolimus, mycophenolate mofetil (MMF), and methylprednisolone. The treatment group was additionally treated with a 16-day course (Day-2 to Day-14) of Cp40, as shown in Fig. 1A. We observed significant prolongation in graft survival with the addition of Cp40, leading to a median graft survival of 15.5 days vs. 4 days ($p = 0.0284$, Fig. 1B).

All primates showed excellent kidney function on the first day after transplant with a minimal rise in serum creatinine (sCr) and blood urea nitrogen (BUN). There was no significant difference between the groups on day 1. However, animals in the control group showed significantly elevated sCr and BUN on day 4 after kidney transplantation and met endpoint criteria, while Cp40-treated animals maintained good graft function (Fig. 1C). We observed no difference in the DSA levels between the two groups postkidney transplantation at these early time points (Fig. 1C and Supplementary Fig. 2A). sCr and BUN for all individual primates throughout the study period are summarized in Supplementary Fig. 3. Additionally, treatment with Cp40 was well tolerated without any treatment-specific side effects. The primates experienced no significant weight loss and were able to maintain their nutritional needs, evident in serum albumin and total protein levels (Supplementary Fig. 4A, B). The addition of Cp40 did not lead to clinically relevant cytomegalovirus (CMV) reactivation, defined as >10,000 viral copy numbers, on daily CMV prophylaxis (Supplementary Fig. 4C). Due to crosstalk between complement and coagulation pathways, we evaluated whether Cp40 treatment promoted coagulopathy. We did not observe any significant impact on blood coagulation measurements, including platelet count, clotting time, D-dimer, etc. (Fig. 1E).

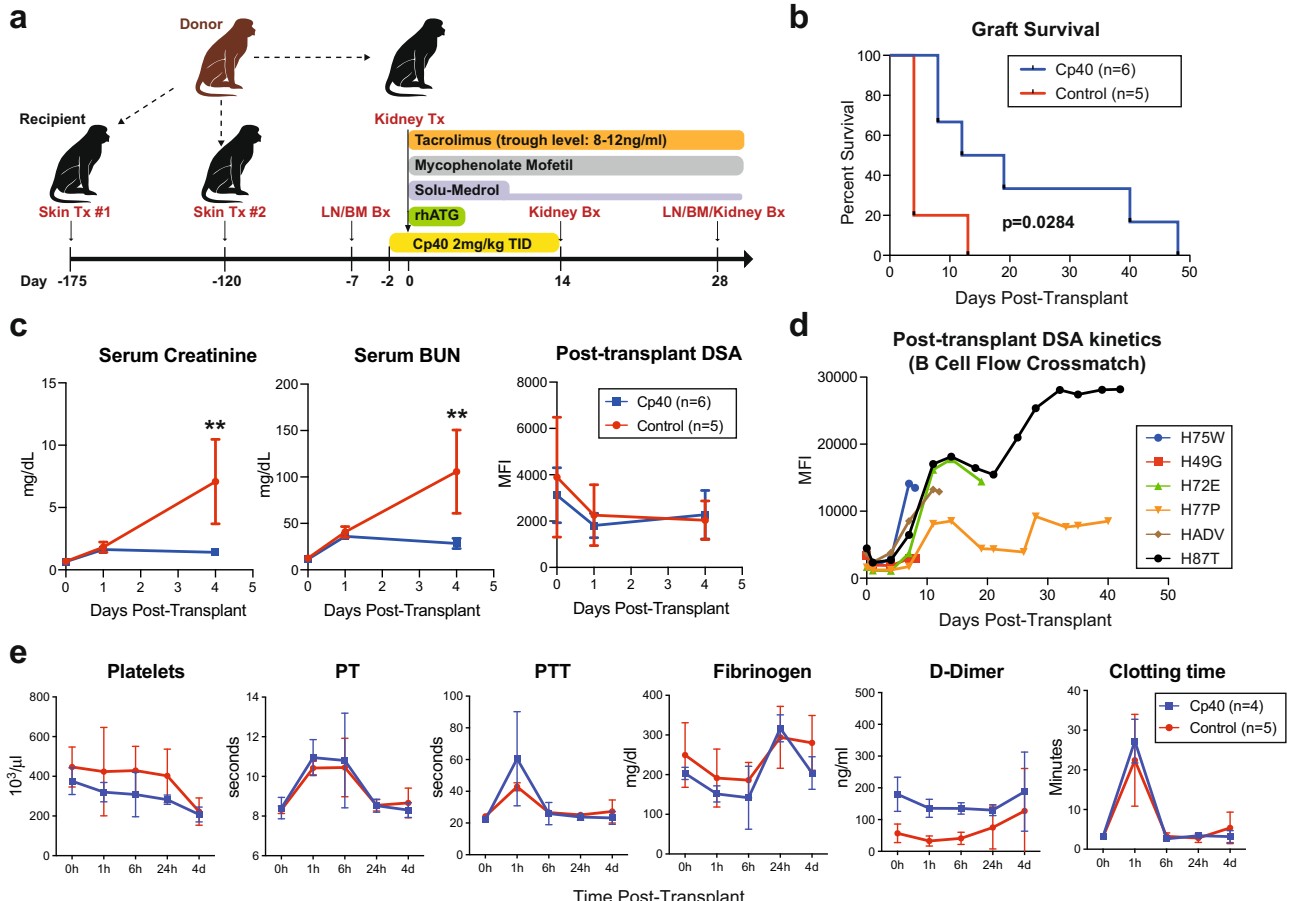

**Fig. 1 Cp40 prevents antibody-mediated rejection despite high levels of donor-specific antibodies. a** A schematic representation of the experimental design and immunosuppression regimen. **b** Kaplan–Meier curve of the graft survival of the control group (red, $n = 5$) and Cp40-treated group (blue, $n = 6$) after life-sustaining kidney allotransplantation. **c** Serum creatinine, blood urea nitrogen (BUN), and circulating DSA (B cell crossmatch) levels of the early post-transplant phase (day 0–4) of the control ($n = 5$) and Cp40 group ($n = 6$). Serum creatinine ($p = 0.0026$) and BUN ($p = 0.0029$) levels were significantly elevated in control animals on post-transplant day 4. Mean ± S.D. is presented. (*$p < 0.05$; and **$p < 0.01$ using two-tailed parametric unpaired $t$ test). **d** Post-transplant circulating DSA of Cp40-treated primates throughout the study period. **e** Post-transplant blood coagulation profiles. Routine lab blood coagulation markers of the control group (red, $n = 5$) and Cp40 group (blue, $n = 4$) during the early post-transplant period (day 0–4. Clotting time was measured by thromboelastography (TEG) during the early post-transplant period (day 0–4). Mean ± S.D. is presented. $N$ number indicates biologically independent animals.

All animals in the treatment group showed a significant increase of DSA after kidney transplantation. Surprisingly, elevated levels of DSA did not lead to immediate graft dysfunction in a subset of animals. Instead, we observed a prolonged period of normal graft function or accommodation after the last administration of Cp40 despite high levels of circulating DSA (Fig. 1D and Supplementary Fig. 2B). These data demonstrate that peri-transplant Cp40 treatment promoted prolonged graft survival without completely mitigating the post-transplant humoral response.

**Cp40 does not alter rhATG-dependent lymphocyte depletion but inhibits lymphocyte activation and proliferation.** We closely monitored the depletion and repopulation of immune cells after kidney transplantation to evaluate possible interference between rhATG and Cp40 since rhATG-mediated lymphocytic depletion could depend on CDC. RhATG caused profound lymphocyte depletion with mean absolute lymphocyte counts (ALC) $< 1 \times 10^3$ cells/μl after the first dose. No significant difference in ALC was observed between the control and treatment groups. Post-transplant T and B cell populations were also

evaluated based on our gating strategy shown in Supplementary Fig. 5. T cell populations, including $CD4^+$ $CD25^+$ FoxP3$^+$ T regulatory cells (Tregs) did not change significantly. We did, however, observe a higher absolute number of $CD20^+$ B cells in the Cp40-treated group postkidney transplantation that reached statistical significance 4 days after transplant ($p < 0.05$, Fig. 2A). After completion of rhATG induction therapy, T cells immediately started repopulating with faster repopulation of $CD8^+$ T cells compared to $CD4^+$ T cells leading to a CD4/CD8 inversion with prolonged and modest CD4 lymphopenia (Supplementary Fig. 6A). We further analyzed the repopulation of T cell subsets by memory and naive phenotype defined by CD28 and CD95. Naive $CD8^+$ T cells showed the most rapid repopulation above baseline within 1 month. In contrast, $CD4^+$ T cells followed repopulation kinetics similar to naive and central memory cells (Supplementary Fig. 6B). RhATG did not lead to significant depletion of innate immune cells, but instead we observed increased circulating level of neutrophils, monocytes, eosinophils, and basophils in Cp40-treated animals (Supplementary Fig. 7). Since complement fragments can impact T and B cell activation and proliferation via complement receptors (e.g., C3aR and C5aR), we further evaluated the lymphocyte profile. Primates

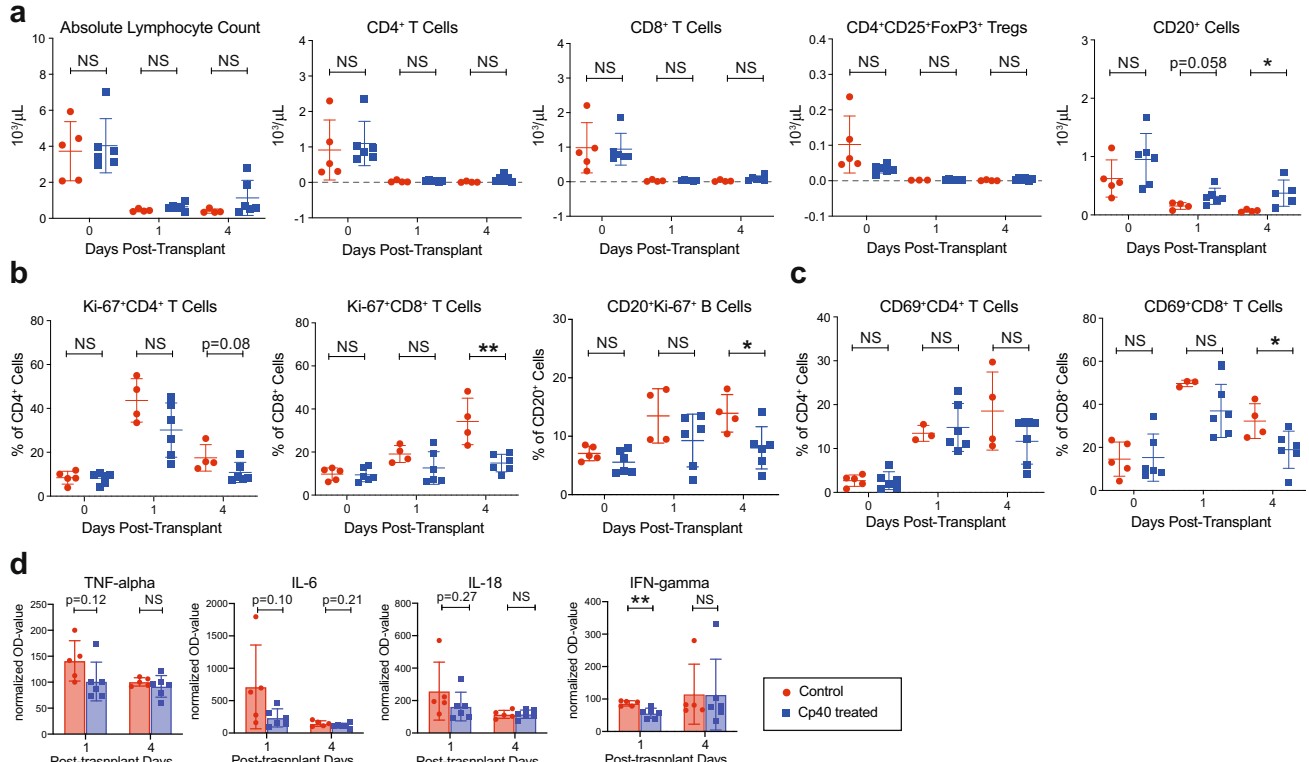

**Fig. 2 Cp40 inhibits lymphocyte activation and proliferation. a** Absolute count of lymphocytes, T cell, T cell subsets including CD4$^+$ CD25$^+$FoxP3$^+$ T regulatory cells, and B cells of the early post-transplant phase (day 0–4) of the control ($n = 5$) and Cp40 group ($n = 6$). At least $n = 3$ was analyzed per time point. **b** Post-transplant T and B cell proliferation in control ($n = 4$ or 5) vs. Cp40 group ($n = 6$). Frequency of proliferating CD4$^+$ T cells, CD8$^+$ T cells, and CD20$^+$ B cells defined through Ki-67 positivity. **c** Post-transplant T cell activation in control vs. Cp40 group. Frequency of activated CD4$^+$ and CD8$^+$ T cells defined through CD69 positivity. **d** Relative changes of post-transplant inflammatory cytokines TNF-α, IL-6, IL-18, and IFN-γ from the control ($n = 5$) and Cp40 group ($n = 6$). All data are presented as mean ± S.D. $N$ number indicates biologically independent animals; *$p < 0.05$; **$p < 0.01$ using two-tailed parametric unpaired $t$ test; NS indicates no statistical significance.

treated with Cp40 showed less T and B cell proliferation, quantified by Ki-67 staining on flow cytometry (Fig. 2B), as well as lower frequency of activated CD69$^+$ CD8$^+$ T cells (Fig. 2C). Furthermore, we measured inflammatory cytokines in the serum after kidney transplantation and observed an early elevation of interferon gamma (IFN-gamma), tumor necrosis factor alpha (TNF-alpha), interleukin 6 (IL-6), and interleukin 18 (IL-18) that were higher in the control group compared to Cp40-treated primates (Fig. 2D). We did not observe any large changes in other cytokines including IL-4 or IL-10 (Supplementary Fig. 8).

**Cp40 prevents early antibody-mediated rejection and inhibits complement deposition in the kidney allograft.** Allografts from control and Cp40-treated animals were evaluated at rejection or at protocol biopsy by an experienced transplant pathologist (A.B.F.) using hematoxylin and eosin (H&E), Periodic acid-Schiff (PAS), and complement deposit staining (Fig. 3A). At the time of graft failure, most animals in the control group had evidence of AMR with a low score of acute cellular rejection (ACR). The subset of primates in the treatment group that experienced early graft failure while on Cp40 treatment (before post-transplant day 14) also showed characteristics of ACR and AMR (Fig. 3B, C). However, in the subgroup of primates that had longer graft survival and a period of accommodation, we did not see any evidence of rejection on POD14 or POD28 protocol biopsies (Fig. 3 and Supplementary Fig. 9). C4d deposition was not indicative of failure of complement inhibition or AMR since C4 is upstream of C3 within the complement cascade. Therefore, we evaluated C3 split product in the graft with

immunohistochemical staining for C3d. In the control group, we were able to show C3d deposition within the kidney graft with predominance in the glomeruli at rejection. One primate in the control group had graft survival of 13 days. Staining of its kidney allograft showed significantly more C3d deposition consistent with a correlation between time and complement deposition (Fig. 3B). Interestingly, the subgroup of animals that experienced early graft failure in the Cp40 group had no significant intragraft C3d deposition despite graft dysfunction and high levels of DSA in the serum. However, after Cp40 was discontinued (POD14), all primates ultimately experienced graft failure and at that time had significant intragraft C3d deposition (Fig. 3B). This suggested that Cp40 blocked the C3 down-stream activation in some animals as reflected by prolonged graft survival.

**Long-term functioning grafts showed downregulation of complement-related genes.** To evaluate the differential outcomes within Cp40-treated animals ($n = 6$), we further divided the treated group based on their rejection timing. Three animals experienced graft failure under Cp40 treatment (early rejecters/ER) while three animals had stable graft function beyond Day 14 (late rejecters/LR). We compared AMR injuries with g + ptc score in these groups. AMR scores were not significantly different between control vs. treated animals, possibly due to the bimodal nature of the treated group (Fig. 4). However, LR kidney histology showed significantly lower AMR scores (Fig. 4A) with less microcirculation inflammation compared to ER (Supplementary Fig. 9A and Supplementary table 3). Interestingly, the level of circulating post-transplant DSA levels/kinetics were not

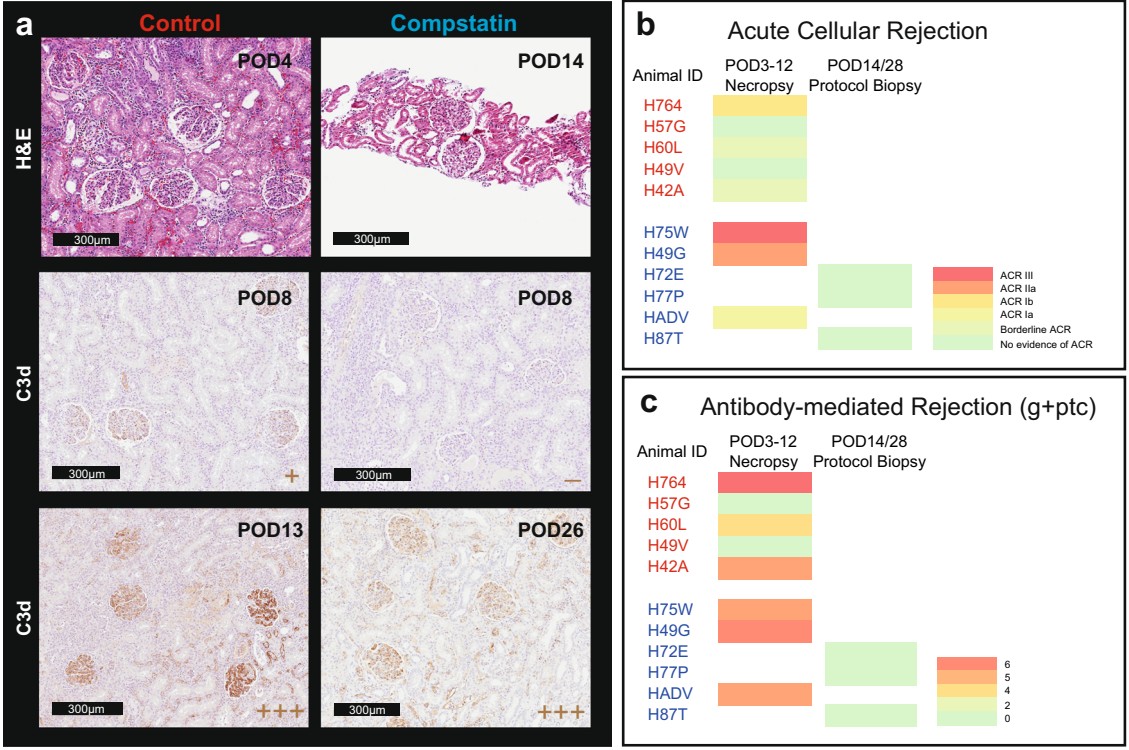

**Fig. 3 Cp40 prevents antibody-mediated graft injury and intragraft C3d deposition. a** Representative images of Hematoxylin and Eosin (H&E) and C3d immunohistochemistry staining of necropsy or biopsy specimens of the control ($n = 5$) and Cp40 ($n = 6$) groups. $N$ number indicates biologically independent animals. **b** Heat map of acute cellular rejection scores of kidney allograft specimens of the control group (top/red) and Cp40 group (bottom/blue). **c** Heat map of acute antibody-mediated rejection scores of kidney allograft specimens of the control group (top/red) and Cp40 group (bottom/blue).

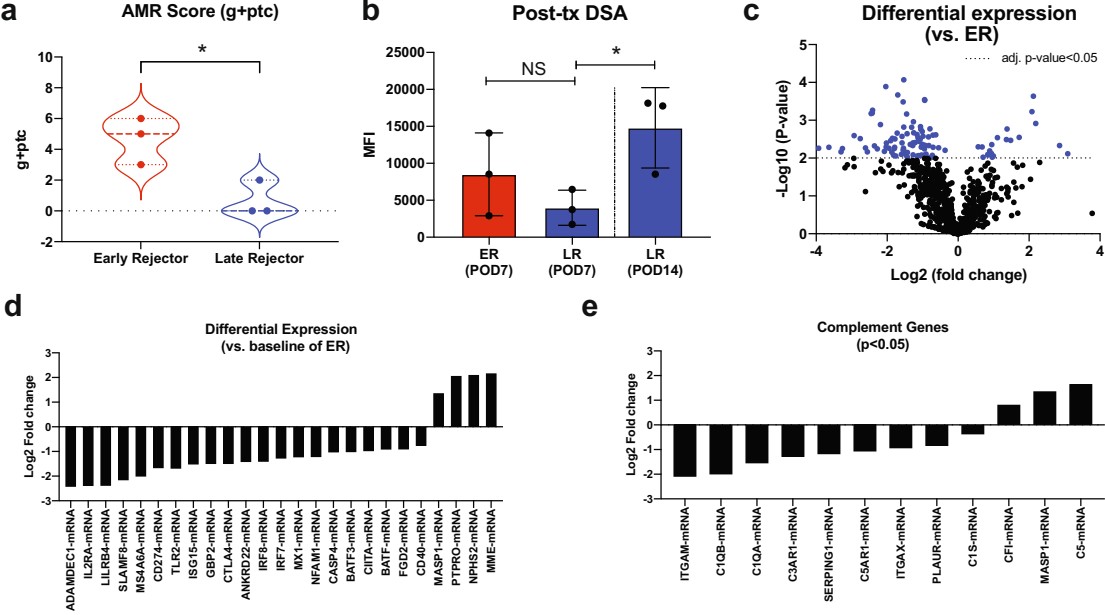

**Fig. 4 Downregulation of complement system related genes promoted long-term graft survival in Cp40-treated animals. a** Antibody-mediated rejection (AMR) score calculated based on Banff grading (g + ptc) from ER ($n = 3$) and LR ($n = 3$) groups. AMR score was significantly elevated in ER group ($p = 0.0224$). Mean ± S.D. is presented. (*$p < 0.05$ using two-tailed unpaired $t$ test). **b** Early post-transplant DSA in ER at day 7 ($n = 3$), LR at day 7 ($n = 3$), and LR at day 14 ($n = 3$). DSA was significantly increased in LR at day 14 compared to LR at day 7 ($p = 0.0365$). Data are presented as mean ± S.D. (NS using two-tailed parametric unpaired $t$ test and *$p < 0.05$ using two-tailed parameteric paired $t$ test). **c** Global gene downregulation in Cp40-treated animals with prolonged graft survival in LR compared to ER. (The calculated $p$-values are adjusted using the Benjamini–Yekutieli method after the regression analysis using a simplified negative binominal model or a log-linear model). **d** Top 25 genes significantly changed in LR compared to ER. **e** Changes in complement system related genes in LR compared to ER. $N$ number indicates biologically independent animals; NS indicates no statistical significance.

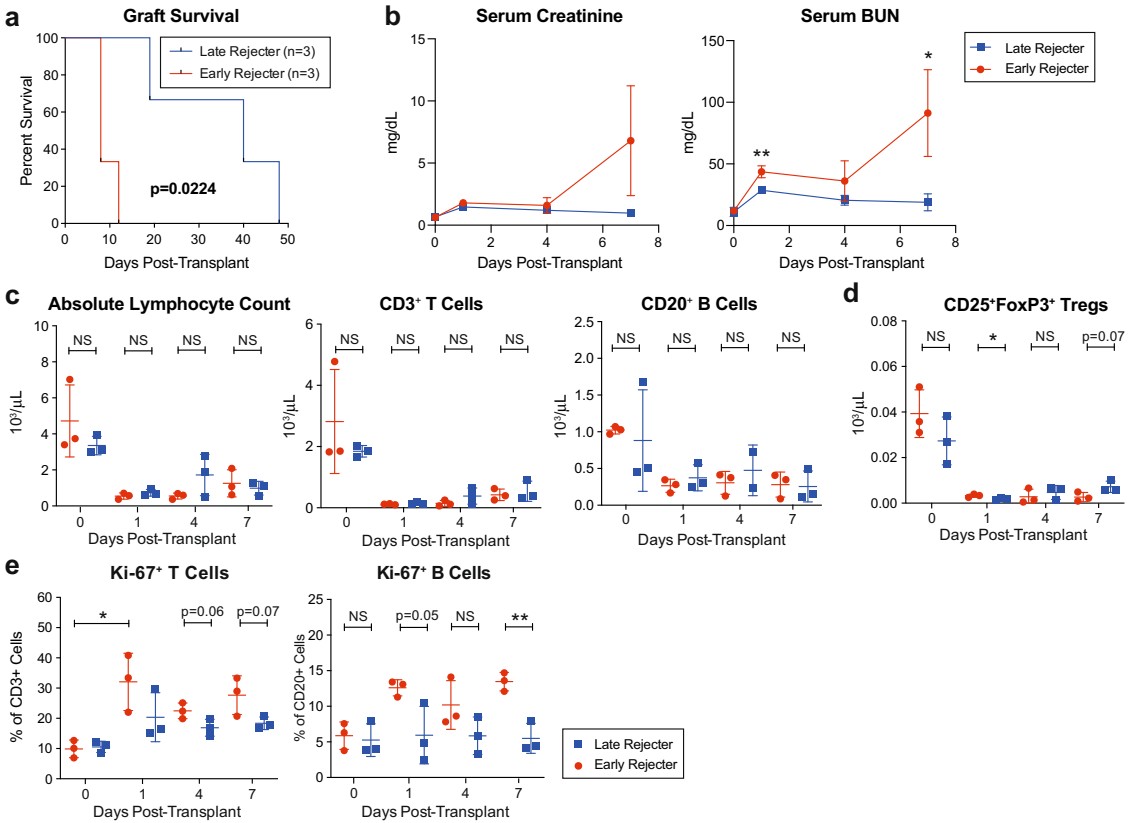

**Fig. 5 A subgroup of Cp40-treated primates experienced graft failure while receiving treatment. a** Kaplan–Meier curve of the graft survival of early rejecters (red, $n = 3$) and late rejecters (blue, $n = 3$). **b** Serum creatinine and blood urea nitrogen (BUN) levels during the first week after kidney transplantation (day 0–7) of the early rejecter group (ER, $n = 3$) and late rejecter group (LR, $n = 3$). BUN level was significantly higher in ER group at post-transplant day 1 ($p = 0.0062$) and 7 ($p = 0.025$) compared to LR group. *$p < 0.05$; **$p < 0.01$ using two-tailed parametric unpaired $t$ test. **c** Absolute lymphocyte, CD3+ T cell and CD20+ B cell counts between the ER ($n = 3$) and LR ($n = 3$) groups. **d** Absolute Treg cell counts defined as CD4+ CD25+ FoxP3+ T cells from ER ($n = 3$) and LR ($n = 3$) groups. Treg cell number was significantly higher in LR group at post-transplant day 1 ($p = 0.0434$) compared to ER group. *$p < 0.05$ using two-tailed parametric unpaired $t$ test; NS indicates no statistical significance. **e** Frequency of Ki-67+ proliferating CD3+ T cells and CD20+ B cells ER ($n = 3$) and LR ($n = 3$) groups. T cell proliferation was significantly increased in ER group (day 0 vs. day 1; $p = 0.0399$). Frequency of proliferating B cells were significantly higher in ER group at post-transplant day 7 ($p = 0.0051$). *$p < 0.05$ using two-tailed parametric paired $t$ test; **$p < 0.01$ using two-tailed parametric unpaired $t$ test; NS indicates no statistical significance. All data are presented as mean ± SD; $N$ number indicates biologically independent animals.

significantly different between ER and LR groups (Supplementary Fig. 9B). Furthermore, circulating DSA levels significantly increased over time without graft rejection (Fig. 4B). Therefore, the alleviation of AMR is not due to a lowered level of DSA in LR group. Even though we observed reduced C3a and C5a plasma levels in CP40-treated animals, the plasma C3a/C5a levels were not different between ER vs. LR. However, graft infiltrating CD68+ macrophages were markedly reduced in LR (Supplemental Fig. 10). To investigate the difference at the molecular level between the ER and LR groups that could conserve the allograft with high DSA levels, we employed gene analysis using a NanoString Platform. RNA samples were prepared from formalin fixed paraffin embedded (FFPE) kidney allograft used for histological evaluation. Interestingly, the pathological difference between the groups was also reflected in the hierarchical clustering analysis. As shown in Supplementary Fig. 11, ER and LR clustered separately and in LR showed that 25 out of 37 pathways were down-modulated, including complement system genes. Similarly, downregulated genes were observed in LR compared to ER (Fig. 4C, D), and 9 out of 12 significantly changed genes related to the complement system were downregulated (Fig. 4E). These data suggest that Cp40 biological activity was clearly different between these two subgroups.

**IgM donor-specific antibodies are associated with early graft and treatment failure**. To investigate reasons for the differential outcome, we further analyzed ER and LR groups. As expected, these two subgroups had a statistically significant difference in graft survival with a median survival of 8 days vs. 40 days ($p = 0.0224$, Fig. 5A). Both subgroups had excellent graft function immediately after kidney transplantation with normal serum creatinine levels. However, the ER group showed an early elevation in serum BUN that was significantly higher than the LR group, suggesting early graft injury (Fig. 5B). Due to the increased morbidity, we did not obtain protocol biopsies within the first week after kidney transplantation to confirm this injury histologically. Both subgroups had good lymphocyte depletion after administration of rhATG, and there was no significant difference in the absolute number of T or B cells throughout the study period (Fig. 5C). In contrast, we observed a higher number of CD4+ CD25+ FoxP3+ Tregs in the LR group 7 days after transplantation ($p < 0.05$, Fig. 5D), which may have contributed to longer graft survival. Additionally, the LR group showed a strong trend towards less T cell proliferation ($p = 0.06$) and significantly less proliferation of B cells ($p < 0.01$, Fig. 5E). These results are consistent with our observation in the control group and suggest that the ER group had incomplete complement

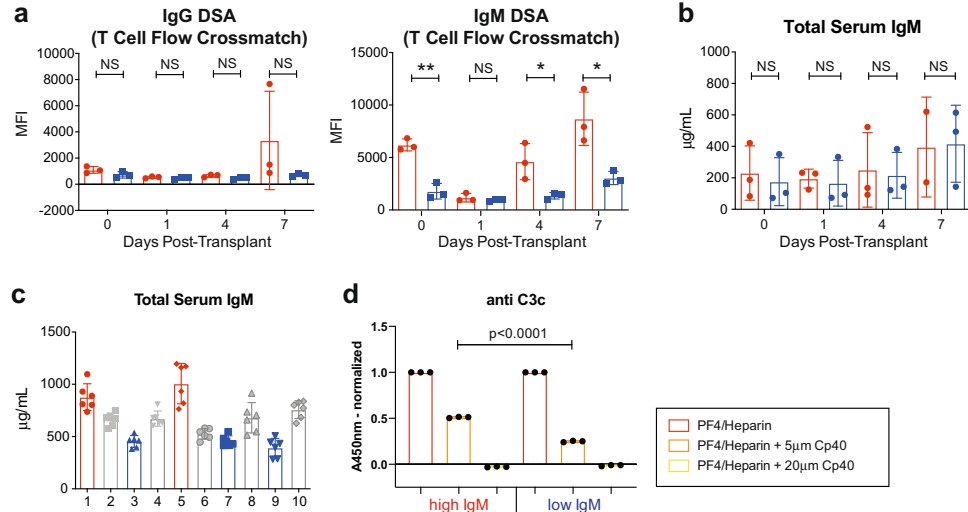

**Fig. 6 IgM DSA is associated with treatment and early graft failure. a** Early post-transplant donor-specific antibody kinetics of IgG and IgM subclass in the early rejecter (ER, $n = 3$) and late rejecter group (LR, $n = 3$). IgM DSA levels were significantly higher in ER group at post-transplant day 0 ($p = 0.0012$), 4 ($p = 0.0322$), and 7 ($p = 0.0203$) compared to LR group. N number indicates samples from biologically independent animals. **b** Total serum IgM levels during the first week post-transplant (day 0–7) in the ER ($n = 3$) and LR ($n = 3$) groups. N number indicates samples from biologically independent animals. **c** Total serum IgM levels of $n = 10$ naive primates not included in this in vivo study. N number indicates samples from biologically independent animals with 6 replicates. **d** In vitro C3c assay comparing the Cp40 treatment effect in primates with high serum IgM levels (red; $n = 3$) and low serum IgM levels (blue; $n = 3$). N number indicates independent experiments. All Data are presented as mean ± SD; *$p < 0.05$; **$p < 0.01$; ***$p < 0.001$ using two-tailed parametric unpaired $t$ test; NS indicates no statistical significance.

inhibition leading to stimulation of T and B cells through circulating complement fragments. Since the differential level of preformed antibody at the time of transplantation could impact susceptibility to complement inhibition, we compared DSA levels between the two subgroups. As shown in Fig. 6A, the level of IgG DSA was similar between the two groups throughout the study period and therefore did not explain the incomplete complement inhibition. Since IgM has the highest potential to activate the classical pathway of the complement cascade, we subsequently measured the IgM DSA levels throughout the study period. The ER group showed significantly higher levels of IgM DSA at the time of transplant as well as postkidney transplantation with an early rebound (Fig. 6A). This observation was specific for the level of donor-specific IgM, as there was no difference in the total serum IgM levels between the groups (Fig. 6B). The presence of IgM DSA was associated with early AMR since we observed no signs of AMR in the group of primates with low levels of IgM DSA. To confirm the IgM-mediated treatment resistance in vitro, 10 additional primates were screened for serum IgM levels. IgM-high and -low primates were tested for complement activation using a well-established PF4/Heparin complex assay. As shown in Fig. 6C, IgM levels were heterogeneous with a wide range of IgM levels (from 392 to 1007 µg/ml) between individual primates. Complement activation was quantified by measuring the C3c concentration. We observed significantly more complement activation in the IgM-high primate, which required higher amounts of Cp40 for complete inhibition ($p < 0.001$, Fig. 6D) consistent with our in vivo results and supporting the hypothesis that the observed treatment failure is due to the presence of IgM DSA, and that the amount of Cp40 used for this study was inadequate to completely inhibit IgM-mediated activation.

## Discussion

In this study, we modeled maximally HLA-incompatible, flow-crossmatch positive kidney allotransplantation in a highly sensitized NHP model. With this model we built an exceptionally high immunologic barrier, leading to accelerated graft rejection in all cases without desensitization. Despite this challenge, we were able to show the ability of C3 complement inhibition to prevent acute AMR. While complement inhibition had no effect on the rebound of DSA after kidney transplantation, we observed less antibody-mediated graft injury and a period of prolonged graft survival in a subset of primates after discontinuation of the C3 inhibitor (Fig. 1). Without interfering with coagulation, Cp40 also inhibited T and B cells activation and proliferation in the lymphopenic environment, highlighting the broad influence of complement inhibition on the immune system beyond alleviating the detrimental effect of DSAs (Figs. 1 and 2). C3 blockade promoted transient stable graft function in the presence of high levels of DSA (accommodation) in some of animals. These responders (LR) showed more down-regulation of immunological pathway-related genes, including complement pathway, compared to the non-responders (ER) (Fig. 4). This was also reflected in longer graft survival, better graft function, and less T and B cell proliferation in animals who responded to C3 blockade (Fig. 5). Surprisingly, retrospective analysis showed that elevated levels of preformed and rebound IgM DSA highly correlated with resistance to C3 blockade in the AMR model (Fig. 6). Finally, we showed that IgM-antigen interaction could promote breakthrough activation of the complement cascade under anti-C3 inhibition.

It is notable that animals treated with Cp40 showed a rapid B cell anamnestic response; however, few animals maintained stable graft function without AMR. No humoral injury in the presence of DSA has been referred to as accommodation[34,35]. This phenomenon has been shown more in ABO-incompatible kidney transplants or in porcine xenografts in NHPs, with incompatible carbohydrate antigens rather than transplantation with HLA-incompatible transplantation[36,37]. Previously, a short course of Yunnan-cobra venom factor, a potent anti-complement protein which showed great efficacy to deplete circulating C3, promoted accommodation, with graft survival of more than 1000 days in conjunction with conventional immunosuppression (CsA, MMF, and steroid) in a skin-sensitized NHP kidney transplant model[38].

These results are truly remarkable but have not been reproduced since. Targeting C3, as the central component of all three complement pathways, is a highly attractive strategy for therapeutic complement inhibition with a conceptual benefit over C1 and C5 inhibition[39]. C1 esterase inhibition only blocks the classical pathway of the complement cascade, and C5 inhibition, as a terminal complement component, allows formation of upstream complement fragments and their participation in activation of the innate and adaptive immune system. The suppression of micro-circulatory inflammation in the presence of preformed and rebounding DSA suggests that the anti-C3 approach alleviates pathogenesis of injury via antibody-dependent cellular cytotoxicity as well. However, even though these animals showed prolonged graft survival, they eventually developed AMR. It is uncertain whether more prolonged targeting of C3 could prevent such gradual development of AMR. Nevertheless, the current study provides a proof of concept that targeting C3 can promote tentative protection of the graft in the sensitized setting without additional desensitization.

Our data also show that Cp40 impacts T cell and B cell activation and proliferation. This could be due to a lower serum concentration of the anaphylatoxins C3a and C5a in the treatment group. The influence of complement fragments on the adaptive immune response and the evidence of C3a and C5a receptors on lymphocytes has been described[40,41]. C3a and C5a have direct effects on T cells through binding of their cognate receptors (C3aR and C5aR) that include effector T cell expansion as well as T cell longevity[42,43]. Anaphylatoxins can further provide costimulation for naive $CD4^+$ T cells[42]. In the absence of C3aR and/or C5aR signaling, T cell-dependent alloimmune responses are dampened and deviated toward Th2 cells and Tregs, instead of Th1[44–46]. Additionally, we observed a dampened pro-inflammatory response with lower levels of IFN-gamma, TNF-alpha, and IL-18 in the Cp40-treated group. Since complement split products, C3a and C5a, are potent chemo-attractants, they may recruit other cellular mediators of inflammation and mediate tissue injury[41,47]. In accordance with this, the graft infiltrating macrophages were greatly reduced in LR kidneys compared to control and ER (Supplementary Fig. 9). However, even though the systemic levels of C3a/C5a were reduced in Cp40-treated animal, we did not observe any differences in plasma C3a/C5a levels between ER and LR. It is possible that the systemic plasma levels do not fully reflect the local concentration of C3a/C5a in a graft. Therefore, we could not directly prove that reduced levels of C3a and C5a are responsible for prolonged graft survival in Cp40-treated animals with inhibited immune responses. Most recently, the C3 inhibitor AMY-101 has been successfully used as treatment for a case of severe COVID-19–associated ARDS, perhaps by a similar mechanism[48].

Even though Cp40 treatment showed biological activity, only some animals fully responded to C3 blockade. We found that preformed IgM confers resistance to Cp40 treatment. The classical pathway (CP) is initiated by plasma C1q binding to the Fc segments of DSA that are bound to HLA antigens[49]. The relative ability of human immunoglobulin to activate the CP is: IgM > IgG3 > IgG1 > IgG2 ≫ IgG4[50]. Secreted IgM complexes in human serum form pentamer or hexamer structures and, when bound to antigens, have the ability to present multiple binding sites for the C1 complex[51]. Therefore, increased IgM against donor antigens could initiate CP very efficiently. Consistent with this explanation, growing clinical observations implicate IgM DSA in transplant rejection. Everly et al. described the association of IgM DSA and graft survival in a cohort of 179 primary renal allograft recipients. The authors were able to show that not the presence of IgM alone but the persistence of IgM with the concomitant presence of IgG3 DSA was associated with significantly shorter graft survival[52]. The presence of IgM alone, however, was associated with higher grades of rejection. A smaller, single institution analysis from the UK that included 92 HLA-incompatible transplant recipients also showed an association between post-transplant IgM levels and graft failure[53]. Importantly, in the setting of sensitized patients undergoing positive crossmatch transplantation, evidence of complement therapy resistance was similarly observed in eculizumab-treated patients with elevated IgM DSA. Three out of 26 patients experienced AMR within the first month after kidney transplantation while on eculizumab treatment, one of them being subclinical rejection. These 3 patients were all found to have elevated levels of IgM DSA while only 1 out of 23 patients in the rejection-free group had detectable levels of IgM DSA[54]. It is also notable that IgM-mediated AMR was shown in highly sensitized patients treated with Imlifidase (IdeS, IgG degrading enzyme) in the absence of circulating IgG DSA[55]. Therefore, IgM DSA may be an independent mediator for initiating AMR. We also show that not the total IgM level but the level of IgM DSA correlates with the incidence of AMR and failure of complement blockade. Furthermore, we were able to show that treatment resistance in primates with high IgM levels can be overcome with an increased dose of complement inhibitor. This suggests that in our in vivo model, an increase in the Cp40 dose may have prolonged graft survival in the high IgM DSA group.

The fact that high-IgM DSA primates experienced graft failure while on Cp40 treatment highlights some of the limitations of this study. All primates received 2 mg/kg Cp40 TID, which in retrospect might be insufficient for controlling the complement cascade in a subset of primates with higher levels of IgM DSA. To translate this regimen into the clinic, more research is required with IgM flow-crossmatch–positive patients to better define the balance between complement activation and inhibition. The goal of this study was to demonstrate efficacy of Cp40 in a very challenging immunologic scenario using serial fully MHC-mismatched skin transplantation and kidney transplantation from the same donor for sensitization. In our experiment, Cp40 treatment alone only achieved short-term protection against AMR. However, similar to previous clinical observations with terminal or proximal complement inhibitors, Cp40 does not completely prevent AMR[56,57]. Since DSAs may continue to injure the allograft despite complete inhibition of the complement system, combined approaches with an Ab-reducing regimen and complement inhibition may promote more stable long-term graft survival in sensitized patients. However, it is also important to note that such a combined approach could increase the risk of infection since C3 has a central role in both opsonization and lysis of infectious microorganisms (i.e. bacteria, virus, and parasite)[58–60]. Therefore, in a human trial we envision complement inhibition being used as an adjunct to (pharmacological) desensitization, and for such a protocol the optimal duration of treatment and risk of infection needs to be defined more clearly. Furthermore, the transient protection/accommodation after complement inhibition could create a unique therapeutic window for both desensitization and AMR treatment. This may facilitate deceased donor kidney transplantation in sensitized recipients by rapidly alleviating the impact of complement-associated AMR injury while the benefit of Ab-reducing regimens takes more time. While Cp40 has not been used in transplantation, a phase I clinical trial has been successfully completed (NCT03316521) and phase II trials are planned.

We have demonstrated the ability of isolated C3 complement inhibition without an antibody-reducing regimen to increase graft survival. The efficacy of Cp40 in mitigating early robust AMR is especially attractive in the following two clinical scenarios: active acute AMR and desensitization prior to deceased donor

transplantation. While further investigations are warranted, we believe that we have demonstrated the considerable potential of C3 complement inhibition that justifies pursuing its translation into clinical transplant applications.

## Methods

**Sensitized nonhuman primate kidney allotransplantation model.** The study animals, male rhesus macaques, were obtained from breeding colonies at Alpha Genesis, Inc. (Yemassee, SC, USA). Maximally MHC class I and II mismatched pairs of NHPs were sensitized to each other with two sequential full-thickness skin transplants[61]. The two skin transplants were performed 8 weeks apart. Sixteen weeks after the second skin transplant, allo-sensitized primate pairs underwent swapping kidney transplantation with contralateral native nephrectomy as previously described[62]. Lymph node biopsies from the axillary and inguinal regions as well as bone marrow biopsies from the iliac crest were performed prior to and 30 days post kidney transplantation. After kidney transplantation, primates were sedated twice per week for physical examination and blood collections. The study endpoint was defined as clinical evidence of acute AMR with worsening kidney function (e.g. rising creatinine, reduced urine output, edema) refractory to methylprednisolone rescue therapy. All animal care and procedures were conducted in accordance with the National Institutes of Health (NIH) guidelines and were approved by the Institutional Animal Care and Use Committee (IACUC) at Duke University (Protocol# A153-18-06).

**Immunosuppressive drug regimens.** All transplanted primates received depletional induction therapy with the rhATG (NIH NHP Reagent Resource, Worcester, MA, USA) at a total dose of 20 mg/kg, administered in 5 evenly divided daily doses between the day of transplant and post-transplant day 4. Maintenance immunosuppression consisted of tacrolimus (Astellas Pharma, Northbrook, IL, USA) IM twice daily (BID), dose adjusted to maintain trough levels at 8–12 ng/mL, MMF (Genentech, San Francisco, CA, USA) 30 mg/kg PO BID, and methylprednisolone (Pfizer, New York, NY, USA) 0.5 mg/kg IM daily starting at day 5 after kidney transplantation after initial taper from 15 mg/kg on the day of transplant. The intervention group was additionally treated with either the compstatin C3 complement inhibitors Cp40[63] or Cp40-KK[64] but not both (the activities of these two molecules are the same), which was given at a dose of 2 mg/kg TID between day −2 prior to kidney transplantation and day 14 after kidney transplantation. A schematic treatment schedule is shown in Fig. 1A. All primates received CMV prophylaxis with 6 mg/kg ganciclovir daily SC throughout the study period. RhCMV viral titers were monitored weekly by polymerase chain reaction as described previously[65]. Clinically suspected ACR episodes (e.g., rise in creatinine, reduced urine output, edema) were treated with methylprednisolone 125 mg/kg IM daily for 3 days, followed by 75 mg/kg IM daily for 3 days, and 25 mg/kg IM daily for 3 days.

**Kidney allograft monitoring and histology.** We assessed the kidney allograft function daily by monitoring urine output and at least twice per week by serum chemistry tests. Protocol percutaneous ultrasound-guided renal biopsies were performed on post-transplant days 14 and 30 as well as at time of suspected AMR. Biopsies were performed with a 20 G CareFusion Coaxial Achieve™ Automatic Biopsy System (BD Biosciences, Franklin Lakes, NJ, USA). We performed standard H&E and PAS staining of all allograft tissue samples. Additionally, C4d, C3d, and CD68 immunohistochemistry was performed for selected biopsy samples and all necropsy specimens. An experienced transplant pathologist (ABF) evaluated and scored the histology specimens according to the Banff criteria[13] in a blinded fashion.

**Blood coagulation studies and thromboelastography.** Citrated blood was collected at 0 h, 1 h, 6 h, 24 h, 4d, and 7d after kidney transplantation. Measurement of the platelet count, D-Dimer, fibrinogen, prothrombin time, and partial prothrombin time was outsourced and measured by (Antech Diagnostics, Fountain Valley, CA, USA) as part of a blood coagulation panel. The activated clotting time was measured by thromboelastography (TEG) as described previously[66]. TEG was performed with 330 μl whole blood and supplementation of 20 μl CaCl as well as 10 μl of Kaolin. All samples were run in duplicate.

**Monitoring of the allogeneic immune response with polychromatic flow cytometry.** Our method for detection of DSA levels in the transplant recipient's serum has been described in detail previously[65]. Briefly, serum samples for analysis were collected throughout the study period. Recipient serum was incubated with donor peripheral blood mononuclear cells (PBMCs). IgG (1:50 dilution) and IgM DSA levels were measured by flow cytometric crossmatch on a BD LSRFortessa™ (BD Biosciences, San Jose, CA, USA) and analyzed using FlowJo software version 10 (Tree Star, Ashland, OR, USA).

Single cell suspension of lymph nodes, bone marrow, and PBMCs were stained for various lymphocyte population markers at the indicated time points. A summary of all fluorochrome-conjugated antibodies is listed in Supplementary table 2. Single cell suspensions were stained with following monoclonal antibodies:

CD4 (L200, dilution 1/100, Catalogue#552838), CD27 (O323, dilution 1/100, Catalogue#46-0279-42), Ki67 (B56, dilution 1/50, Catalogue#561284 and 556027), CD45RA (L48, dilution 1/50, Catalogue#347723), FoxP3 (259D, dilution 1/50, Catalogue#320212), IgD (polyclonal, dilution 1/200, Catalogue#2030-02), CCR7 (150503, dilution 1/20, Catalogue#561143), IgG (G18-145, dilution 1/20, Catalogue#561296), CD20 (2H7, 1/20, Catalogue#560631 and 302314), CD25 (CD25-3G10, dilution 1/20, Catalogue#MHCD2505 and 170-081-029), CD8 (RPA-T8, dilution 1/100, Catalogue#557760 and 558207), CD127 (eBioRDR5, 1/100, Catalogue#17-1278-42), CD19 (CB19, dilution 1/5, Catalogue#ab197060), PD-1 (eBioJ105, dilution1/50, Catalogue#17-2799-42), CD3 (SP34-2, dilution 1/100, Catalogue#560770). The flow cytometry was performed on a BD LSRFortessa™ (BD Biosciences, San Jose, CA, USA) and analyzed using FlowJo software version 10 (Tree Star, Ashland, OR, USA). Our flow cytometry gating strategies are summarized in Supplementary Fig. 5.

**Cytokine multiplex immunoassay.** Serum samples were collected from all primates through the study period. Serum levels of inflammatory cytokines (GM-CSF; IFN gamma; IL-1 beta; IL-10; IL-12p70; IL-13; IL-17A; IL-18; IL-2; IL-23; IL-4; IL-5; IL-6; TNF alpha) were measured in the early post-transplant phase (POD0-7) using the Th 14-Plex NHP ProcartaPlex Panel (Invitrogen, Waltham, MA, USA).

**Plasma C3a and C5a ELISA.** The concentration of plasma C3a and C5a was quantitated by enzyme-linked immunosorbent assay (ELISA) using Monkey C3a ELISA Kit (AssayGenie, Dublin, Ireland) and Monkey C5a ELISA Kit (MyBioSource, Sandiego, CA) according to the manufacturers' instructions. The ELISA plates provided had been pre-coated with capture antibodies specific to Complement Component 3a or Complement Component 5a. 100 μL of standard or plasma samples (dilution of 1:3) were added and incubated for 90 min at 37 °C. Samples were diluted with sample diluent to a total volume of 100 μL in each well. The standard curves were generated with the recombinant C3a or C5a starting at a concentration of 10 ng/mL (serial dilation as 10, 5, 2.5, 1.25, 0.625, 0.3125, 0.15625, 0 ng/mL as suggested). ELISA plates were washed three times with PBST and 100 μL of biotinylated detection antibody were added and incubated for 1 h at 37 °C. After washing the plate three times with PBST, 100 μL of HRP conjugated working solution was added to each well and incubated for 30 min at 37 °C. ELISA plates were then washed five times with PBST and developed by adding 90 μL of substrate reagent to each well at room temperature approximately for 15 min, stopped with 50uL of stop solution. The plates were read at 450 nm.

**Gene expression analysis using the NanoString Platform.** We isolated total RNA from FFPE kidney allograft blocks using a recently published technique by Adams et al.[67]. Samples were included in the final analysis if they passed a priori quality control criteria (260:280 ratio > 1.7). We measured gene expression by using the NanoString nCounter MAX platform (NanoString Technologies, Seattle, WA). We used the nCounter Human Organ Transplantation Panel (No. LBL-10743-01) to measure 758 genes covering the core pathways and process surrounding host response and rejection of transplanted tissues (including 12 internal reference genes for data normalization). Gene expression data were log2-transformed, background subtracted, and normalized to the geometric mean expression of 12 housekeeping genes by using the nSolver software platform 4.0 (Nanostring Technologies).

**Assessment of NHP serum antibody levels and in vitro PF4/heparin assay.** We measured total IgM (Behtyl Human IgM ELISA Quantitation Set, Montgomery, TX, USA) serum levels in all primates at various time points as indicated in the results with ELISA. To measure complement activation and inhibition in vitro, plasma was first incubated with PF4/heparin complexes (25 μg/ml and 0.25 U/mL respectively; formed at a PF4 to heparin molar ratio (PHR) of 6.6). After 1 h of incubation, complement-fixed antigen was captured by KKO, a PF4/heparin specific monoclonal antibody, and complement fragments containing C3 were detected using a biotinylated anti-C3c antibody as previously described[68].

**Statistical analysis.** Statistical analyses were performed using GraphPad Prism software version 5.0 (GraphPad Software, San Diego, CA, USA). Survival data were plotted using the Kaplan–Meier method and log-rank test was performed to determine statistical significance. Statistical comparisons between different groups were performed using the student $t$ test and values of $p < 0.05$ were considered statistically significant.

**Reporting summary.** Further information on research design is available in the Nature Research Reporting Summary linked to this article.

## Data availability

The primary data that support the findings of this study are available from the corresponding author upon reasonable request. Supplementary information accompanies this paper and the nanostring data have been deposited in NCBI's Gene Expression Omnibus (GEO) and are accessible through GEO Series accession number GSE178843. Source data are provided with this paper.

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

## Acknowledgements

This work was supported by the National Institute of Allergy and Infectious Diseases of the National Institutes of Health as part of the NHP Transplantation Tolerance Cooperative Study Group under the U19AI131471 (awarded to S.J.K.) and P01-AI068730 (awarded to J.D.L.). The content is solely the responsibility of the authors and does not necessarily represent the official views of the National Institutes of Health. Rhesus-specific anti-thymocyte globulin (rhATG) used in this study was provided by the NIH Nonhuman Primate Reagent Resource (R24 OD010976, U24 AI126683).

## Author contributions

R.S. performed experiments, analyzed data, and wrote the manuscript. P.M.S., Z.W.F., A.Y.C., and M.M. participated in N.H.P. procedures and contributed to data interpretation as well as review of the manuscript. J.Y., M.S., J.S.Y., S.K., and G.M.A. performed in vitro experiments. A.B.F. reviewed and graded renal allograft histology. E.S.R. and J.D.L. helped design the N.H.P. experiment, performed in vitro assays, and contributed to data interpretation. J.K. and S.J.K. designed the experiments, performed the transplants, interpreted the data, and contributed significantly to writing the manuscript.

## Competing interests

J.D.L. is the founder of Amyndas Pharmaceuticals, which is developing complement inhibitors for therapeutic purposes and inventor of patents or patent applications that describe the use of complement inhibitors for therapeutic purposes, some of which are developed by Amyndas Pharmaceuticals. J.D.L. is also the inventor of the compstatin technology licensed to Apellis Pharmaceuticals (i.e., 4(1MeW)7 W/POT-4/APL-1 and PEGylated derivatives such as APL-2/Pegcetacoplan). The other authors declare no competing interest.
