## [Peer Review File · Nature Communications]

REVIEWER COMMENTS

Reviewer #1 (Remarks to the Author):

This paper by Schmitz et al describes the use of a novel C3 complement inhibitor Cp40 in a primate model of allo-sensitization and antibody rejection. The group is well established in this area and the model has important characteristics regarding translation to management of highly-HLA sensitized patients. This is a very interesting model and certainly important in terms of developing rational therapies for desensitization of highly HLA sensitized patients. As the authors point out desensitization is still a work in progress and that the use of plasmapheresis IVIG and rituximab have limitations in antibody removal to limit HLAi transplantation and prevent antibody rebound. The authors propose the use of a novel C3 inhibitor Cp40 which inhibits all 3 complement activation pathways that could be of benefit in preventing early immunologic injury to the allograft and potentially modify immune responses to induce a state of accommodation. The authors show that transient treatment of sensitized primates prolonged median allograft survival from 4->15 days, despite persistence of DSAs. 50% of Cp40 treated animals maintained graft function beyond the last day of treatment. There were reductions in allograft injury markers post-treatment.

This paper is of interest due to its examination of a novel agent with possible applications to HLAi transplantation in humans. However, the findings are of concern since the effect is modest. In fact, similar observations have been made with eculizumab in patients transplanted with crossmatch positivity. Here, the titer of DSAs is very important in that those with CDC + often have only transient or no protection from antibody rejection and graft loss is more likely. Similar findings have been seen with C1INH with some beneficial effects that would be up that appear to be similar to what they are seeing in the primate model. Of interest is a paper by Bohmig (Eskandary F, Jilma B, Mühlbacher J, Wahrmann M, Regele H, Kozakowski N, Firbas C, Panicker S, Parry GC, Gilbert JC, Halloran PF, Böhmig GA. Anti-C1s monoclonal antibody BIVV009 in late antibody-mediated kidney allograft rejection-results from a first-in-patient phase 1 trial. *Am J Transplant*. 2018 Apr; 18(4): 916-926.) examined the use of an anti-C1s monoclonal for treatment of ABMR in humans where pre-and post treatment measurements of complement activation parameters, DSAs, biopsy BANFF scores and molecular microscope analysis of ABMR associated genes was accomplished. These authors found that the anti-C1s completely inhibited C3 activation and resulted in marked reductions in post-treatment C4d in allograft biopsies. However, no change in microcirculation inflammation, gene expression patterns, DSA levels, or kidney function was found. These findings suggest that DSAs are important and continue to injure the allograft despite complete inhibition of the complement system. This is likely through ADCC pathways. Here antibody reduction therapies are critical for the success of any desensitization protocol. Another interesting point of this paper is the modification of B-cell responses in the Cp40 treated animals. The authors indicate that this is likely due to reduced levels of C3a and C5a which can upregulate both T- and B-cell responses. It would be of interest to determine if reductions in the C3a and C5a levels are responsible for inhibition of these immune responses in this system. This is also important since other cellular mediators of inflammation (macrophages and PMNs) avidly respond to C5a and are likely important in mediation of allograft injury. Despite the observed limitations of complement inhibition alone, the study is very interesting does explore the use of a novel complement inhibitor however the impact of this treatment is modest in terms of improving graft survival and certainly not sufficient to allow transition to the clinic at this time. It would be of interest to see how effective the agent is when coupled with antibody reduction therapy and whether there can be some long-term benefits observed in this very important model.

Reviewer #2 (Remarks to the Author):

Overall, distal (C5) and proximal (C1) complement inhibition studies have been disappointing because of limited efficacy, reproducibility, or both. This study provides an exciting proof of concept that C3 complement inhibition mitigates antibody mediated injury to the kidney allograft in a rigorous sensitization non-human primate model. The investigators demonstrate that 3/6 animals treated with C3 inhibition had long graft survival, with significant inhibition of circulating

IgM DSA, and hypothesize that treatment failure in the remaining 3 animals was due to under-dosing. The conclusions are supported by the findings, although the sample size is rather small. Nevertheless, there is strong proof of concept that C3 inhibition may be an important tool to help prevention and treatment strategies for ABMR. The following clarifications can strengthen the conclusions of this study:

-Treatment clarification: Did animals in the intervention arm receive Cp40, Cp40-KK, or both?

-FIG1-D. It would be useful to provide a panel B cell FXM between the two groups (Median Channel Shifts for example)

-FIG3 (renal pathology). Images are not convincing and C3d is not a standard biomarker of allograft injury. Please provide images showing microcirculation inflammation (ptc + g). Please provide data on intragraft monocyte/macrophages which express complement receptors and play an important role in ABMR

-Cytokines IL6, IL18, and TNF-alpha were not significantly different between the two groups, making the discussion on Th2 phenotype challenging. Please provide a comment or data on IL4, IFN-gamma and IL10 regulatory cytokines

-Please report p values consistently (NS or 0.1, 0.3, etc)

Point-by-point response for NCOMMS-20-41725

RE: Manuscript ID NCOMMS-20-41725 entitled "C3 Complement Inhibition Promotes Transient Accommodation and Prolongs Renal Allograft Survival in Sensitized Non-human Primates".

We thank the reviewers for the thoughtful review of our manuscript and we are pleased that our manuscript is potentially acceptable for publication in the *Nature Communications*. We are grateful for the opportunity to provide a detailed response to the reviewers' queries and comments. Our point-by-point response is provided below and the revisions incorporated in the revised text are highlighted:

Robin Schmitz
Jean Kwun
Stuart Knechtle

REVIEWER COMMENTS

Reviewer #1 (Remarks to the Author):

This paper by Schmitz et al describes the use of a novel C3 complement inhibitor Cp40 in a primate model of allo-sensitization and antibody rejection. The group is well established in this area and the model has important characteristics regarding translation to management of highly-HLA sensitized patients. This is a very interesting model and certainly important in terms of developing rational therapies for desensitization of highly HLA sensitized patients. As the authors point out desensitization is still a work in progress and that the use of plasmapheresis IVIG and rituximab have limitations in antibody removal to limit HLAi transplantation and prevent antibody rebound. The authors propose the use of a novel C3 inhibitor Cp40 which inhibits all 3 complement activation pathways that could be of benefit in preventing early immunologic injury to the allograft and potentially modify immune responses to induce a state of accommodation. The authors show that transient treatment of sensitized primates prolonged median allograft survival from 4->15 days, despite persistence of DSAs. 50% of Cp40 treated animals maintained graft function beyond the last day of treatment. There were reductions in allograft injury markers post-treatment.

This paper is of interest due to its examination of a novel agent with possible applications to HLAi transplantation in humans. However, the findings are of concern since the effect is modest. In fact, similar observations have been made with eculizumab in patients transplanted with crossmatch positivity. Here, the titer of DSAs is very important in that those with CDC + often have only transient or no protection from antibody rejection and graft loss is more likely. Similar findings have been seen with C1INH with some beneficial effects that would be up that appear to be similar to what they are seeing in the primate model. Of interest is a paper by Bohmig (Eskandary F, Jilma B, Mühlbacher J, Wahrmann M, Regele H, Kozakowski N, Firbas C, Panicker S, Parry GC, Gilbert JC, Halloran PF, Böhmig GA. Anti-C1s monoclonal antibody BIVV009 in late antibody-mediated kidney allograft rejection-results from a first-in-patient phase 1 trial. *Am J Transplant*. 2018 Apr;18(4):916-926.) examined the use of an anti-C1s monoclonal for treatment of ABMR in humans where pre-and post treatment measurements of complement activation parameters, DSAs, biopsy BANFF scores and molecular microscope analysis of ABMR associated genes was accomplished. These authors found that the anti-C1s completely inhibited C3 activation and resulted in marked reductions in post-treatment C4d in allograft biopsies. However, no change in microcirculation inflammation, gene expression patterns, DSA levels, or kidney function was found. These findings suggest that DSAs are important and continue to injure the allograft despite complete inhibition of the complement system. This is likely through ADCC pathways. Here antibody reduction therapies are critical for the success of any desensitization protocol. Another interesting point

of this paper is the modification of B-cell responses in the Cp40 treated animals. The authors indicate that this is likely due to reduced levels of C3a and C5a which can upregulate both T- and B-cell responses.

-It would be of interest to determine if reductions in the C3a and C5a levels are responsible for inhibition of these immune responses in this system.

: We evaluated plasma C3a and C5a levels as suggested. As shown in the figure on the right, we can identify a strong trend of reduced systemic C3a and C5a in Cp40 treated animals compared to animals without Cp40 (controls). This might simply reflect the systemic effect of anti-C3 similar to what was shown by Mastellos et al (ref#48). However, we do not see any difference between ER and LR (see figure below). This lack of difference is possibly due to the low experimental numbers, but it is also possible that the systemic plasma level of complement split products simply does not accurately reflect the local/intragraft concentration. Therefore, we were unable to demonstrate that reduced levels of C3a and C5a are responsible for prolonged graft survival in Cp40 treated animals. We mentioned these observations in the text; however we did not add them as a figure since the reduction C3a/C5a with Cp40 treatment has been previously described. Furthermore, the sample quality has not been equally preserved for control and CP40-treated group since there was a time gap between setting up these two groups. The control samples were potentially exposed to more freeze-thaw cycles. Finally, as mentioned above, we cannot attribute the differential survival to the plasma C3a/C5a levels.

- This is also important since other cellular mediators of inflammation (macrophages and PMNs) avidly respond to C5a and are likely important in mediation of allograft injury.

: This is a great point. We evaluated the levels of monocytes/macrophages in the secondary lymphoid organs and the graft. The level of (CD14+) monocytes was not different among groups (control vs. Cp40 or ER vs. LR). However, interestingly, the level of graft infiltrating (CD68+) macrophages was clearly lowered in LR group compared to control or ER. This might reflect the lower levels of local chemoattractants (i.e. C3a and C5a) in long-term surviving (LR) animals. However, there are numerous potential reasons for this difference. Since we cannot identify a reduction of C3a/C5a in the graft, we cannot attribute the monocyte/macrophage changes to C3a/C5a. We added this information as **Supplementary Fig. 10**.

Supplementary Figure 10

A

B

Despite the observed limitations of complement inhibition alone, the study is very interesting does explore the use of a novel complement inhibitor however the impact of this treatment is modest in terms of improving graft survival and certainly not sufficient to allow transition to the clinic at this time.

- It would be of interest to see how effective the agent is when coupled with antibody reduction therapy and whether there can be some long-term benefits observed in this very important model.

: Yes, this is an excellent point and we completely agree with reviewer#1's opinion that antibody reduction therapies are critical. Some animals (LR group) showed significantly prolonged graft survival; however, they eventually rejected the kidney allografts by AMR. Combined approaches with Ab reduction and and complement inhibition may promote more stable long-term graft survival in sensitized patients. Furthermore, the transient protection/accommodation after complement inhibitor could create a unique therapeutic window for both desensitization and AMR treatment. This might allow deceased donor kidney grafts to be done in sensitized patients and rapidly alleviate on-going AMR, whereas the effect of an Ab-reducing regimen takes time. We added this point in the discussion.

Reviewer #2 (Remarks to the Author):

Overall, distal (C5) and proximal (C1) complement inhibition studies have been disappointing because of limited efficacy, reproducibility, or both. This study provides an exciting proof of concept that C3 complement inhibition mitigates antibody mediated injury to the kidney allograft in a rigorous sensitization non-human primate model. The investigators demonstrate that 3/6 animals treated with C3 inhibition had long graft survival, with significant inhibition of circulating IgM DSA, and hypothesize that treatment failure in the remaining 3 animals was due to under-dosing. The conclusions are supported by the findings, although the sample size is rather small. Nevertheless, there is strong proof of concept that C3 inhibition may be an important tool to help prevention and treatment strategies for ABMR. The following clarifications can strengthen the conclusions of this study:

- Treatment clarification: Did animals in the intervention arm receive Cp40, Cp40-KK, or both?

: Some animals received Cp40 and others Cp40-KK but not both. It was unrelated to their survival outcome. The activities of these two molecules are the same (Berger et al, J Med Chem, 2018; <https://pubmed.ncbi.nlm.nih.gov/29920096/>). We clarified this in the main text.

-FIG1-D. It would be useful to provide a panel B cell FXM between the two groups (Median Channel Shifts for example)

: The rapid DSA rebound was observed in Cp40 treated animals. The rebound kinetics are more apparent compared to the control group because the graft survival was prolonged. Early post-transplant DSA kinetics were presented in Fig. 1C (BFXM) and in Supplementary Fig. 2A (TFXM). We evaluated if there is any difference in DSA rebound in ER vs. LR which could impact the rejection kinetics and survival. As expected, we do not find any evidence of differential levels of DSA between these two groups. As shown in figure 2B, LR group showed continuously increased DSA over time which suggests that the level of DSA is not the deciding factor for graft survival in CP40 treated animals. We added DSA levels of ER and LR groups in the text and added as a Supplementary Fig. 8A as below.

-FIG3 (renal pathology). Images are not convincing and C3d is not a standard biomarker of allograft injury. Please provide images showing microcirculation inflammation (ptc + g). Please provide data on intragraft monocyte/macrophages which express complement receptors and play an important role in ABMR

: We now provide representative images showing microcirculation inflammation in ER and LR. As shown in the figure below, H75W (ER, POD8) had glomerulitis (green arrow) and peritubular capillaritis (blue arrows) that was essentially absent in H77P (LR, POD14). This is now added as supplementary fig. 8B.

For intragraft monocyte/macrophages, please see our answers to reviewer#1. Additionally, in the revised version of the manuscript we added full BANFF grading in supplementary table 3 (see below).

Supplemental Table 3: Banff Scoring of kidney allograft from Cp40 treated animals

Group	Case	POD	i	iv	ii	iii	IFTA	g	ci	ct	cs	mm	cv	sh	ptc	Diagnosis
-------	------	-----	---	----	----	-----	------	---	----	----	----	----	----	----	-----	-----------

ER	H75W	8	3	8	2	2	8	9	1	0	0	0	0	3	ACR, type 3 (focal arterial fibrinoid necrosis with mural inflammation); findings also suspicious for AMR.	
ER	H49G	8	1	1	2	2	8	9	1	1	19	3	3	0	2	ACR, type 2A; Findings also suspicious for AMR
ER	HADV	12	3	3	2	2	0	3	1	1	19	0	1	0	0	ACR, type 3 (focal arterial fibrinoid necrosis with mural inflammation & aneurysmal-type change). Findings are at least suspicious for AMR.
LR	H87T	28	1	0	0	0	0	0	3	1	0	0	0	0	2	"Borderline changes"; Findings also suspicious for AMR (tubular injury & peritubular capillaritis)
LR	H77P	14	0	0	0	0	0	0	0	0	0	0	0	0	0	No evidence of ACR. Mild tubular injury.
LR	H72E	14	0	0	0	0	0	0	0	0	0	0	0	0	0	No evidence of ACR. Mild tubular injury.

ACR = acute cellular rejection, AMR = antibody-mediated rejection.

-Cytokines IL6, IL18, and TNF-alpha were not significantly different between the two groups, making the discussion on Th2 phenotype challenging. Please provide a comment or data on IL4, IFN-gamma and IL10 regulatory cytokines

We did not observe any significant or strong trend of reduction or increase of serum IL-4 and IL-10 in Cp40 treated animals compared to controls. Interestingly, serum IFN-g at POD1 was significantly reduced in Cp40-treated animals. We added IFN-gamma data in figure 2D as shown below.

Additionally, we evaluated serum cytokine levels with ER and LR separately. Even though there might be a trend of reduced cytokine in LR, it is hard to claim this due to low N numbers. We added this information as Supplementary Fig. 8. As shown below, we did not observe any large changes to other cytokines including IL-4 or IL-10.

-Please report p values consistently (NS or 0.1, 0.3, etc)

: We are now reporting p-value consistently as follows through the manuscript. NS, not significant, * <0.05, ** <0.01, ***<0.001.

REVIEWERS' COMMENTS

Reviewer #1 (Remarks to the Author):

This is a re-submission of a manuscript detailing the impact of C3 inhibition with the novel C3 activation inhibitor Cp40. The investigators examined the impact of C3 inhibition on IgM DSAs, graft survival, Banff scores and immune cell activation post-transplant. The authors appear to have improved the manuscript with additional data that clarified points of concern. I do think they should discuss the limitations of their observations and clarify points of concern about C3 inhibition alone as a desensitization strategy. This could be done by referencing and discussing previous studies mentioned in the first review in the discussion section. This is certainly a very interesting drug that needs further vetting in animal models before human studies are undertaken in highly sensitized patients. I also think it would be important to address issues of infection risk since inhibition of the alternative pathway would likely expose patients to high risk of bacterial infection.

Reviewer #2 (Remarks to the Author):

My concerns have been addressed. The main issue is that some animals received cp40 whereas others received cp40-kk, but the study does not separate the two groups because the mechanism of action is similar. Similar mechanism of action does not imply similar effects e.g. different potency of depleting monoclonal antibodies against CD20. I would recommend adding this in the limitation of the study.

Response to Referees Letter_NCOMMS-20-41725B

RE: Manuscript ID NCOMMS-20-41725 entitled "C3 Complement Inhibition Promotes Transient Accommodation and Prolongs Renal Allograft Survival in sensitized Non-human Primates".

We thank the referees and editor for the thoughtful review of our manuscript and we are pleased that our manuscript is accepted in principal for publication in the *Nature Communications*. We are grateful for the opportunity to provide a detailed response to the reviewers' queries and comments. Our response to referees is provided below and the revisions incorporated in the revised text are highlighted:

Jean Kwun
Stuart Knechtle

REVIEWERS' COMMENTS

Reviewer #1 (Remarks to the Author):

This is a re-submission of a manuscript detailing the impact of C3 inhibition with the novel C3 activation inhibitor Cp40. The investigators examined the impact of C3 inhibition on IgM DSAs, graft survival, Banff scores and immune cell activation post-transplant. The authors appear to have improved the manuscript with additional data that clarified points of concern. I do think they should discuss the limitations of their observations and clarify points of concern about C3 inhibition alone as a desensitization strategy. This could be done by referencing and discussing previous studies mentioned in the first review in the discussion section. This is certainly a very interesting drug that needs further vetting in animal models before human studies are undertaken in highly sensitized patients. I also think it would be important to address issues of infection risk since inhibition of the alternative pathway would likely expose patients to high risk of bacterial infection.

: We now incorporated the limitation of C3 inhibition alone as a desensitization strategy and risk of infection with combinatorial approach in discussion as "However, similar to previous clinical observations with terminal or proximal complement inhibitors, Cp40 does not completely prevent AMR. Since DSAs may continue to injure the allograft despite complete inhibition of the complement system, combined approaches with an Ab-reducing regimen and complement inhibition may promote more stable long-term graft survival in sensitized patients. However, it is also important to note that such a combined approach could increase the risk of infection since C3 has a central role in both opsonization and lysis of infectious microorganisms (i.e. bacteria, virus, and parasite)". The revised portion is highlighted in the text.

Reviewer #2 (Remarks to the Author):

My concerns have been addressed. The main issue is that some animals received cp40 whereas others received cp40-kk, but the study does not separate the two groups because the mechanism of action is similar. Similar mechanism of action does not imply similar effects e.g. different potency of depleting monoclonal antibodies against CD20. I would recommend adding this in the limitation of the study.

: Cp40-KK is Cp40 with 2 Lys at the C-terminus to increase solubility. As previously reported, the activities of these two molecules are the same (Berger et al, J Med Chem, 2018; <https://pubmed.ncbi.nlm.nih.gov/29920096/>). We clarified this in the main text.